# Effect of Thermal Treatment of Aluminum Core-Shell Particles on Their Oxidation Kinetics in Water for Hydrogen Production

**DOI:** 10.3390/ma14216493

**Published:** 2021-10-29

**Authors:** Olesya A. Buryakovskaya, Mikhail S. Vlaskin, Anatoly V. Grigorenko

**Affiliations:** Laboratory of Energy Storage Substances, Joint Institute for High Temperatures of the Russian Academy of Sciences, 125412 Moscow, Russia; presley1@mail.ru

**Keywords:** hydrogen, core-shell particles, aluminum, thermal treatment, oxidation kinetics

## Abstract

The effect of thermal treatment of aluminum core-shell particles on their oxidation kinetics in water for hydrogen production was investigated. The samples were obtained by dividing dried aluminum powder, partially oxidized by distilled water, into eight portions, which were thermally treated at temperatures of 120, 200, 300, 400, 450, 500, 550 and 600 °C. Alumina shell cracking at 500–600 °C enhances hydrogen generation due to uncovering of the aluminum cores, while sharp thickening of the protective oxide film on the uncovered aluminum surfaces at 550–600 °C significantly reduces reactivity of the core-shell particles. For these reasons, after reaction with distilled water at 90 °C for two hours, the highest hydrogen yield (11.59 ± 1.20)% was obtained for the sample thermally treated at 500 °C , while the yield for aluminum core-shell powder without heat treatment was only (5.46 ± 0.13)%. Another set of experiments employed multiple consecutive cycles of alternating oxidation by water and thermal treatment at 500 °C of the same powder sample. As predicted, the hydrogen yield gradually decreased with each subsequent experiment. The series of six cycles resulted in a total hydrogen yield of 53.46%.

## 1. Introduction

Until quite recently the implementation of clean energy technologies has seemed almost exotic. However, we now observe an accelerating evolution of the conventional energy industry based on fossil fuels towards new environment-friendly concepts. One of these concepts is to use carbon-neutral energy carriers, such as hydrogen, that can be consumed without greenhouse gas emissions and produced without a carbon footprint. Other reasons for the high degree of attention to hydrogen include its high gravimetric energy density and the availability of methods that ensure highly efficient conversion of hydrogen energy [1]. These high-efficiency energy conversion methods include hydrogen utilization in fuel cells, or its combustion in a combustion chamber to drive gas and steam turbines with expanding heated air or steam, produced from water heating by the exhausted air, respectively [2].

Presently, there are a variety of approaches to hydrogen production either from fossils or from renewable sources (e.g., biomass, water) [3]. Most hydrogen is generated by natural gas reforming and coal gasification processes which are emissions-intensive; biomass is not currently considered as a source for industrial-scale hydrogen production, while water represents a promising alternative. The established methods of water splitting, which are currently attracting much attention, include thermolysis, photolysis, radiolysis, and electrolysis [3,4]. Small-scale production of ‘green’ hydrogen from water can be achieved by its reaction with so-called ‘metal-based energy carriers’, such as magnesium and aluminum. This approach has the following advantages: it provides a feasible method of producing pure hydrogen without its separation from other gases, it is not accompanied by harmful gas emissions, and it does not require expensive large-scale equipment or complicated technology.

The application of aluminum and magnesium to obtain hydrogen is also anticipated to be profitable in terms of waste management, which is an important current problem. There are many methods for the utilization of aluminum and magnesium scraps for hydrogen production, some of which are described in [5,6,7,8,9,10]. It is also important to note that in 2018, ‘Alcoa’ and ‘Rio Tinto’ (well-known aluminum manufacturers) announced their ‘ELYSIS’ project applying revolutionary smelting technology, based on inert anodes, that produces oxygen and eliminates greenhouse gases [11,12]. Aluminum is, therefore, anticipated to become ‘green’ providing a further argument in support of its use for hydrogen generation. Since aluminum is a multi-purpose metal, widely used in industry, a variety of aluminum-based composite materials are known. Aluminum is used for the production of high-purity γ-Al_2_O_3_ [13] and α-Al_2_O_3_ (the raw material for leucosapphire crystals) [14], porous alumina catalyst carriers [15], and production of aluminum-based core-shell powders reinforced with Al_2_O_3_, which represent promising materials for the fast-growing additive manufacturing industry [16] due to its enhanced mechanical and physical properties [17,18,19]. These composite materials can be manufactured via aluminum oxidation by pure water resulting in the production of a pure solid reaction product. Moreover, hydrogen generated during the oxidation process can be converted into useful energy, thus contributing to the efficiency of the manufacturing process in terms of energy saving, another goal to be pursued.

Under standard conditions (room temperature, atmospheric pressure), the reaction between pure water and magnesium or aluminum (excepting that in the form of nanopowders) can barely proceed because of a thin dense oxide layer on the surfaces of these metals, or because of the fast formation of limited permeability, low-soluble layers of the reaction products (e.g., Mg(OH)_2_, AlOOH, Al(OH)_3_), onto them at the very beginning of the process. The reaction therefore requires either higher temperatures [20,21,22] or activation procedures to destroy the protective layers. A number of activation methods for their disruption, dissolution or removal in other ways are known. For example, tested techniques for magnesium include: ball-milling of magnesium powders together with chlorine salts (AlCl_3_, NaCl, KCl) to obtain hydrogen from pure water [23,24], preparation of hydroreactive ‘mechanical alloys’ of magnesium and other metals (Ni, Fe, Co, Cu) [25,26,27], and implementation of a Pt-coated titanium net or stainless steel net as a catalyst for magnesium alloy ingots to promote hydrogen evolution from conductive (NaCl) aqueous solution [28,29]. The list of suggested methods for aluminum activation is even more extensive. Thus, aqueous solutions of salts (e.g., CoCl_2_ or NiCl_2_ [30]), or alkali (e.g., KOH or NaOH) or acid (HCl) solutions [31,32,33], have proved to be effective for obtaining hydrogen from aluminum in dispersed or bulk forms at moderate or even below-zero temperatures. The addition of different metals to aluminum, for instance, low-melting-point metals, such as Ga, In, Mg, Li, Zn, Bi, Sn, or metals with higher melting points, such as Ca and Cu) [34,35,36,37,38,39,40], or iron group metals, such as Fe, Co, Ni [41] (alone or with additives such as organic fluoride, NaCl, SnCl_2_, Bi_2_O_3_), by alloying or ball-milling, also enhances aluminum reactivity. Aluminum oxidation can be promoted as well by its ball-milling with various additives, such as metal oxides, water soluble salts, and other compounds, such as chlorides of metals, carbon nitride, etc. [42,43]. The reaction of aluminum with water can be facilitated by aluminum compounds including aluminum hydroxide (gibbsite, bayerite), boehmite [44,45,46,47,48,49], and alumina-based catalysts, as they can be employed, for instance, to promote the production of aluminum-based core-shell powders. No addition of other compounds, potentially resulting in product contamination, is needed. Even if no composite materials are to be produced, the reaction product composed of aluminum hydroxides, aluminum oxides, or their mixture (and, probably, some amount of unreacted aluminum) can be used for aluminum regeneration without a need for separation from impurities. Techniques for manufacturing composite materials of aluminum and alumina include aluminum powder ball-milling together with α- or γ-Al_2_O_3_ powders [50,51,52,53], aluminum ball-milling with Al(OH)_3_ powder with subsequent sintering in vacuum at 600 °C resulting in the formation of γ-Al_2_O_3_ phase [54], and aluminum mixing with γ-Al_2_O_3_ powder with its sintering in vacuum at 600 °C [55].

Summarizing the above, in previous studies, it has been clearly demonstrated that aluminum is a promising material for hydrogen production, and its reaction with water can be effectively promoted by different additives including aluminum hydroxide and alumina. In [56] a process for manufacturing aluminum-alumina composites via the oxidation of aluminum powder by water at 120–200 °C with hydrogen co-production is demonstrated. The present study is intended to investigate the effect of the thermal treatment of aluminum core-shell powder at different temperatures on its oxidation kinetics in water for hydrogen production at a moderate temperature of 90 °C, and to determine whether a substantial degree of aluminum conversion (e.g., hydrogen yield) can be achieved.

## 2. Materials and Methods

### 2.1. Original Reagents

In the present study, the original reagents were a fine aluminum powder ‘APZh’ (TU 1791 99-024-99) and distilled water. The average aluminum powder particle size was determined from grain size measurements which were carried out using a laser light scattering particle size analyzer, Analysette 22 (Fritsch GmbH, Idar-Oberstein, Germany). The particle size distribution histogram is given in Figure 1. The average size of the powder particles was 34.68 µm.

To investigate the surface structure of the powder particles, scanning electron microscopy (SEM) analysis with a field emission scanning electron microscope, model JSM 7401F (JEOL Ltd., Tokyo, Japan) was performed.

The SEM micrographs of the aluminum powder are shown in Figure 2. The aluminum particles represented on the micrographs had an approximately spherical shape.

The aluminum powder composition was determined via an x-ray fluorescent spectrometer Thermo Scientific Niton XL3t GOLDD+ XRF Analyzer (Thermo Fisher Scientific Inc., Billerica, MA, USA). The results are represented in Table 1.

### 2.2. Experimental Plant

In the experiments on aluminum powder oxidation in hot water the experimental facility shown in Figure 3 was employed. The reactor with a cooling jacket has its inlet and outlet connected to a thermostat. The thermostat is used to maintain a constant temperature in the reactor during an experiment. The temperature of the media in the reactor is measured with a resistance temperature detector. The reactor is installed on a magnetic mixer. Hydrogen obtained during an experiment is supplied through silicon hoses to a gas meter. The readings of the resistance temperature detector and the gas meter are transferred to a computer for on-line visualization.

### 2.3. Experimental Procedure

In each experiment, the reactor was filled with 400 mL of distilled water and aluminum powder was added to it. Then the reactor was sealed and the media inside it (initially at room temperature) were heated up to 90 °C. This temperature was maintained at a constant level during each experiment.

In the first experiment, aluminum powder (25.0752 g) was added to water and after 2 h the suspension was discharged, subjected to decantation, and then the resulting partially oxidized aluminum powder was dried in a drying oven at 120 °C (3 h). The powder was then divided into eight parts each of which was used as the original material in the following experiments. The first portion was used as it was and the other seven parts were passed through thermal treatment in a muffle furnace at different temperatures (200, 300, 400, 450, 500, 550 and 600 °C, 2 h). Hydrogen yields were measured via the gas meter.

## 3. Results

### 3.1. Effect of Temperature

The first set of experiments was carried out to determine the effect of the thermal treatment temperature on the degree of aluminum conversion (hydrogen yield) of different samples. Firstly, a sample of original aluminum powder (‘primary’ sample with a mass of 25.0752 g) was reacted with distilled water at 90 °C for two hours. Then the resulting solid product (partially oxidized aluminum powder) was separated from water and dried in a drying oven at 120 °C. The dried solid product (composed of powder particles containing both unreacted aluminum and aluminum oxidation product) was then divided into eight samples (‘subsamples’ or ‘secondary’ samples with approximately equal masses), each of which was individually thermally treated. Seven of the eight samples underwent thermal treatment at 200, 300, 400, 450, 500, 550 and 600 °C (2 h of holding at constant temperature); the eighth was not thermally treated in any way other than drying at 120 °C (in practice, drying was common to all eight ‘secondary’ samples, as they were ‘derived’ from the same ‘primary’ sample).

Each of the eight samples was in turn added to distilled water, and the suspension was heated to 90 °C (the duration of all experiments was 2 h). The corresponding curves of hydrogen yield (represented as aluminum conversion degree per sample gram) vs. time for the original sample, and the eight samples produced from it, are shown in Figure 4. The ‘secondary’ samples contained a reduced amount of aluminum (as compared to the original sample). The curves for the ‘secondary’ samples are represented as ‘extensions’ of the curve for the ‘primary’ (original) sample in order to demonstrate the reduction of the unreacted aluminum content in the samples.

For each of the ‘original’ powder samples and ‘secondary’ samples thermally treated at 300 °C, the measured values were averaged over three experiments, and for each of the ‘secondary’ samples thermally treated at 400, 500 and 600 °C, the obtained data sets were averaged over two experiments. For the repeated tests the standard deviation values were calculated, which are represented in Figure 4 and in Table 2. As can be seen, the data sets for the ‘original’ powder sample and for the ‘secondary’ sample thermally treated at 600 °C show quite good repeatability; the deviation ranges for the ‘secondary’ samples after thermal treatment at 400 °C and 500 °C overlap, while results for the sample thermally treated at 300 °C demonstrate a large uncertainty in the experimental data.

The hydrogen production curves demonstrate that the thermal treatment of partially oxidized aluminum samples at temperatures of 400, 450, 500 and 550 °C considerably enhanced their reactivity with water as compared to the original sample of aluminum powder and the samples thermally treated at 200, 300 and 600 °C, as well as the sample without thermal treatment in a muffle oven (passed only through drying at 120 °C). The corresponding conversion degrees (hydrogen yields) for the samples are given in Table 2 (the total conversion degrees account for the contribution of the original sample, i.e., 5.46%, while the conversion degrees per experiment account for only the amount of aluminum converted into hydrogen during the experiments with the ‘secondary’ samples). The probable reasons for such a difference in the standard deviations for the various samples are suggested below, following discussion of the X-ray diffraction analysis results).

The highest conversion degree (hydrogen yield) corresponded to the sample thermally treated at 500 °C, (11.59 ± 1.20)% during an experiment (17.05% in total). This experimental value was almost twice as high as the equivalent value for the original aluminum powder, (5.46 ± 0.13)%. The data in Table 2 clearly demonstrate that heat treatment at 200, 300 and 600 °C, as well as the absence of heat treatment in a muffle oven (that is equal to heat treatment at 120 °C in a drying oven), provided a quite low degree of conversion; the amount of hydrogen produced during 2 h of experiment approximately halved in comparison with that for the original powder. Thermal treatment in the temperature range from 400 °C to 550 °C significantly improved the reactivity of the samples compared to both the original sample and the samples thermally treated at the other temperatures. All the curves started to rise after about 20 min from the beginning of the experiments; this time corresponds to the heating of the suspension in the reactor (in all the experiments traceable hydrogen production started at a temperature of ~68–70 °C). Almost all the curves have a section with a high reaction rate, and after about 20–40 min reaction rates became almost constant. Such reaction slowdown may be attributed to the formation of fresh aluminum hydroxide aggregations on the surfaces of the powder particles.

Figure 5 shows the corresponding micrographs of the thermally treated samples before the experiments. The particles comprise aluminum cores covered with the shells of the reaction product. As can be seen, the shells on the samples thermally treated at 500 °C and 600 °C have cracks, and the shells on the particles thermally treated at 600 °C appear to have shrunk compared with those for the samples thermally treated at 500 °C.

As the results show, the samples thermally treated at 120, 200 and 300 °C were composed of aluminum, boehmite and bayerite. The samples treated at 400 °C and higher contained alumina, which agrees with the results obtained in other studies. For example, experimental data on oxide-film growth for pure aluminum reported in [57] showed that at temperatures of 673 K (400 °C) and higher the amorphous oxide film becomes crystalline Al_2_O_3_, and a novel model described in [58] predicts formation of Al_2_O_3_ at temperatures over 578 K (305 °C), and formation of Al(OH)_3_ and AlOOH at lower temperatures. As the corresponding lines for some components of the powder were too close to each other, in some cases precise identification of components was hindered (the corresponding peaks are marked as two listed components). With respect to the crystalline form of aluminum oxide, it is more likely that in such doubtful instances it was still γ-Al_2.144_O_3.2_ rather than η-(Al_2_O_3_)_1.333_.

The structure of the reaction product had substantially the same appearance for all the samples. Figure 6 shows the reaction product layer for the sample thermally treated at 300 °C. As can be seen, the product mostly formed a flake-like structure. There were also relatively large crystals on the samples which probably appeared during the reaction due to crystallization of the product dissolved in water.

The results of X ray diffraction analysis are given in Figure 7.

The X-ray diffraction plots demonstrate the formation of crystalline aluminum oxide phases (η-(Al_2_O_3_)_1.333_ or γ-Al_2.144_O_3.2_) at temperatures of 400 °C and higher and, at the same time, a considerable increase in the degree of aluminum conversion was observed for 400 °C compared to that for the preceding tested thermal treatment temperature of 300 °C. Such an effect can presumably be attributed to a higher concentration of Lewis acid sites for alumina in comparison with those for boehmite, as a lower Al atom coordination generally corresponds to a stronger Lewis acidity, and the susceptibility of Lewis acid sites to water is well-known [59,60,61]. The experiments described in [55] demonstrated that the mere presence of γ-Al_2_O_3_ promotes aluminum oxidation in pure water even without thermal treatment, which supports the view that it has a catalytic effect. A decrease in the aluminum conversion degree when passing from 400 °C to 450 °C is likely to have the same explanation, i.e., the Lewis acidity of η-Al_2_O_3_ is higher than that of γ-Al_2_O_3_ [62,63].

Considering the standard divergences for different samples, their relatively high values for the samples thermally treated at 300, 400 and 500 °C can be attributed to random damage to some particle shells composed of boehmite, bayerite or aluminum oxide phases, as a result of mechanical effects (impacts, friction, etc.) occurring during the manipulations with the powder samples. With respect to the sample thermally treated at 600 °C, its alumina shells are perhaps more dense and tight due to sintering (from the corresponding micrograph in Figure 5, it can be seen that shrinkage of the shells took place) and therefore was more resistant to random mechanical damages; the original powder simply did not contain shells of aluminum-water reaction products.

When summarizing the data from the micrographs, it might be expected that the samples thermally treated at a temperature of 500 °C and higher would react with water more rapidly than other samples, due to the formation of cracks in the shells of alumina causing uncovering of the aluminum cores. However, in practice, thermal treatment at the higher temperatures (550 and 600 °C) resulted in lower rates of aluminum oxidation.

According to the XPS analysis results reported in [57], the thickness of the passive oxide film on the aluminum surface was drastically affected by temperature. For the aluminum substrate heated under temperatures of 100–300 °C for 15,000s it was less than 1 nm, while at temperatures of 400 °C and 500 °C, it increased to 3.9 nm and 6.1 nm, respectively, for the same exposure time. However, these values are still relatively low; according to Figure 1.2, ‘Growth of oxide skin’, represented in the *‘Handbook of Aluminium Recycling’*, Part III, Section 1.2, ‘Oxidation during melting’, p. 87 [64], a discontinuity in the oxidation trend occurs, expressed as a sudden and sharp increase in film thickness (from nearly 6 nm at 500 °C up to approximately 20–21 nm at 600 °C). Such an effect is ascribed to the transformation of an amorphous oxide structure to a crystalline structure.

In order to check whether the observed decrease in the reaction rate can be attributed to the formation of a thicker oxide layer on the uncovered surfaces of aluminum cores at temperatures above 500 °C, a sample of unreacted aluminum powder was analyzed using a thermogravimetric (TG) method. The analysis was performed using a thermogravimetric analyzer STA PT1600 (Linseis Messgeraete GmbH, Selb, Germany). The sample was heated from room temperature (23 °C) to 600 °C in air atmosphere at a flow rate of 30 mL/min, and a heating rate of 5 °C/min. During the analysis, the relative mass change of the powder sample, from the reaction between aluminum and oxygen contained in the air flow resulting in the formation of aluminum oxide on the particles surfaces, was measured. The obtained TG curve is represented in Figure 8.

The TG curve is steepest between approximately 500 and 600 °C (levels of 500, 550 and 600 °C are marked with dashed lines). Within this range, the relative mass change (attributed to the formation of a passive aluminum oxide layer on the powder particle surfaces) rose from 0.24 to 0.83%. This result is consistent with the formation of a considerably thicker passive oxide layer at temperatures over 500 °C.

Therefore, 500 °C is the optimum temperature value tested in this study. At this temperature, crack formation in alumina shells provides aluminum core uncovering, and water contacts aluminum cores through these cracks, and the oxide film forming on the uncovered areas of aluminum cores (as thermal treatment in a muffle furnace proceeded under air atmosphere) is relatively thin. At the higher tested temperatures (550 and 600 °C) the oxide film forming on the uncovered aluminum surfaces becomes too thick (and its hydration in water preceding the start of the aluminum core oxidation is much slower). At lower temperatures (400 and, presumably, 450 °C) considerable crack formation does not occur and, therefore, the surface layer separating aluminum cores and water is much thicker.

In [52,54] the samples of aluminum and γ-Al_2_O_3_ thermally treated at 600 °C provided much better results in terms of hydrogen production than the samples of pure aluminum powder, while in the present study an opposite effect occurred. This difference can be attributed to the fact that, in the other studies, γ-Al_2_O_3_ was ball-milled with aluminum resulting in the formation of inclusions on the surface of the aluminum particles and thermal treatment took place in a vacuum. In contrast, in the present study, no mechanical treatment was employed and thermal treatment proceeded in air atmosphere, favorable for the formation of a thick layer of passive oxide film on the surface areas of aluminum cores uncovered due to the cracking of the aluminum oxide shells.

### 3.2. Effect of Multiple Alternating Oxidation and Thermal Treatment

The second set of experiments was intended to investigate the effect of multiple cycles of alternating oxidation and thermal treatment at 500 °C (with preliminary drying at 120 °C) of the powder sample obtained in the first set of experiments under the same conditions (drying at 120 °C and thermal treatment at 500 °C) on its aluminum conversion degree (hydrogen yield). As in the previous series of experiments, partially oxidized aluminum powder was reacted with distilled water heated up to 90 °C for 2 h (2 h and 8.5 min for the first powder oxidation after treatment). The resulting curves of aluminum conversion degree (per sample gram) vs. time are represented in Figure 9. In the experiments, the same sample was tested, the amount of aluminum in it gradually decreased. The curves corresponding to the experiments of this series are represented as successive ‘extensions’ of each other and the curve for the ‘primary’ (original) sample (this enables illustration of the reduction of aluminum content in the sample due to its conversion into hydrogen and solid reaction product).

The corresponding conversion degrees (hydrogen yields) for the sample are given in Table 3 (total conversion degrees account for the contributions of the initial and all preceding experiments, while conversion degrees per experiment account for only the amount of aluminum converted into hydrogen during the corresponding experiment).

Figure 9 and Table 3 clearly demonstrate that the degree of aluminum conversion (per experiment) of the sample gradually became lower with each subsequent experiment involving thermal treatment, as expected. This trend can be explained by a decrease in the sizes of the aluminum cores of the partially oxidized aluminum powder resulting in the reduction of the interface area between aluminum and water. However, up to and including the fifth cycle, the degree of conversion was higher than that for the original sample of aluminum powder, (5.46 ± 0.13)%. The total conversion degree achieved in this experiment series was 53.46%.

In all the experiments, two-thirds of the hydrogen yields obtained over 2 h were produced in 35–50 min of the start (or 15–30 min from the beginning of the reaction, as the reaction starts at approximately 68–70 °C), and all the curves demonstrated reaction deceleration caused by the formation of the product layer on the powder particle surfaces.

As shown in the micrographs (see Figure 5 and Figure 6), there are two major structure types formed by the solid product: a flake-like spongy structure and bulk crystalline structures. As the bulk crystalline structures form due to the dissolution and further crystallization of the product on the crystal nuclei, an increase in their amount and sizes with an increase in the reaction time might be expected. Therefore, to obtain aluminum core-shell powder with a higher uniformity of product distribution on the particle surfaces and its structure, it would be recommended to reduce the duration of reaction between aluminum and water in each cycle of successive thermal treatment and oxidation.

In summary, successive alternating procedures (thermal treatment at 500 °C and oxidation by water), with a reduced reaction period, were tested for co-production of hydrogen and composite material containing aluminum cores and alumina shells with a flake-like spongy structure. This approach has potential application to the manufacture of composite products with high accuracy in the control of the aluminum and alumina contents during the reaction via hydrogen yield measurement. Hydrogen co-production and its effective utilization are further potential advantages of the process.

## 4. Conclusions

In the present study the effect of thermal treatment of aluminum core-shell powder on its oxidation kinetics in water for hydrogen production was studied.

The first set of experiments sought to determine the effect of thermal treatment temperature on hydrogen yield. For this purpose, the original sample of aluminum powder was reacted with distilled water and heated up to 90 °C for two hours, then it was dried and divided into eight samples (seven samples thermally treated at 200, 300, 400, 450, 500, 550 and 600 °C in a muffle oven and one sample passed only through drying at 120 °C), each of which was then reacted with distilled water heated up to 90 °C for two hours. Heat treatment at 200, 300 and 600 °C, as well as the absence of heat treatment (sample after drying at 120 °C), resulted in relatively low conversion degrees, while heat treatment from 400 to 550 °C provided higher conversion degrees. The highest aluminum conversion degree (hydrogen yield) corresponded to the sample thermally treated at 500 °C, (11.59 ± 1.20)% during the experiment (and 17.05% in total), i.e., almost twice as high as the same value for the original powder, (5.46 ± 0.13)%. This can be ascribed to crack formation in the alumina shells with aluminum uncovering and a relatively thin oxide film forming on the uncovered areas of aluminum cores, as compared to treatment at higher temperatures.

The second set of experiments was focused on the impact on hydrogen production performance of multiple cycles of alternating oxidation and thermal treatment at 500 °C of the powder sample. Aluminum conversion degree (hydrogen yield per experiment) gradually became lower with each subsequent treatment, as expected. However, up to and including the fifth treatment, the conversion degree was higher than that for the original sample of aluminum powder. The total conversion degree (hydrogen yield) achieved in this experimental series (six experiments with thermally treated powder) was 53.46%.

The successive cycles of the alternating procedures (thermal treatment at 500 °C and oxidation by water) described in the present paper can potentially provide a useful technique for the co-production of hydrogen and composite material containing aluminum cores and alumina shells with desired contents of aluminum and alumina (which can be controlled during the reaction with high accuracy by hydrogen yield measurement).

## Figures and Tables

**Figure 1 materials-14-06493-f001:**
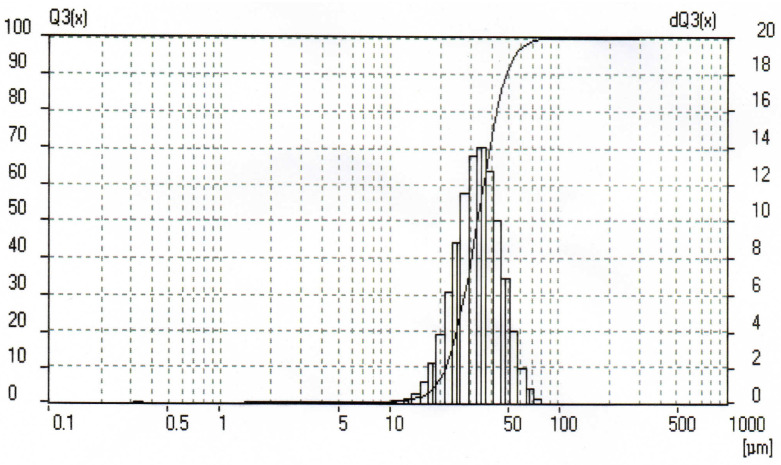
Aluminum powder particle size distribution histogram.

**Figure 2 materials-14-06493-f002:**
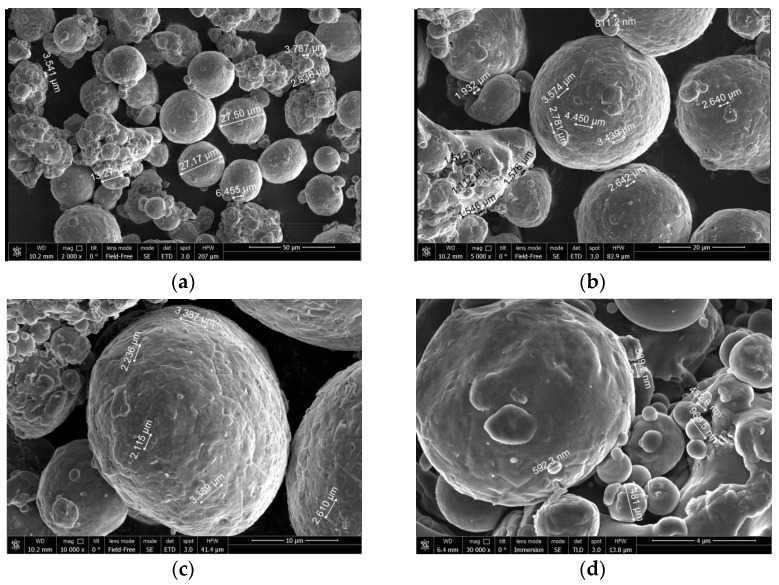
SEM micrographs of magnesium powder. Scale: (**a**) 50 µm, (**b**) 20 µm, (**c**) 10 µm, (**d**) 4 µm.

**Figure 3 materials-14-06493-f003:**
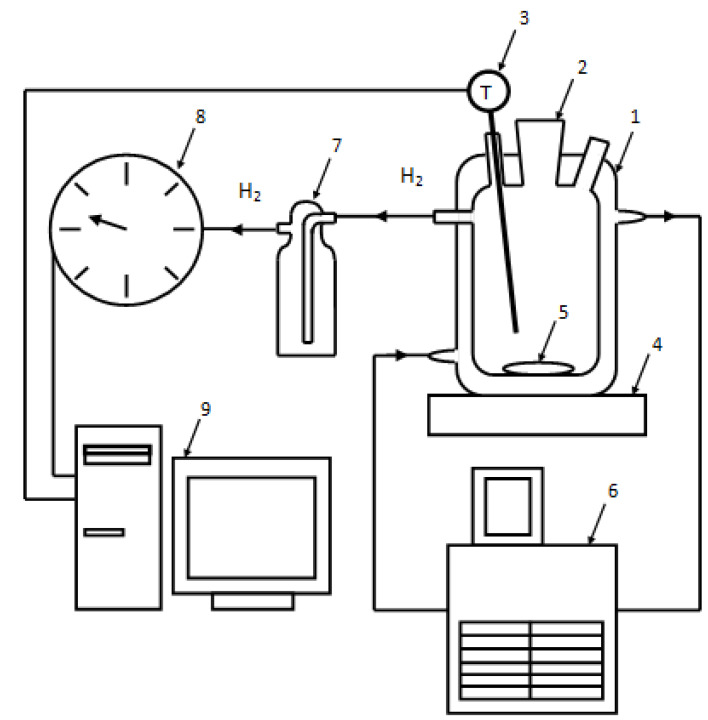
Experimental plant: 1—reactor, 2—filling nozzle, 3—resistance temperature detector, 4—magnetic mixer, 5—stir bar, 6—thermostat, 7—Drexel flask, 8—drum-type gas meter, 9—computer.

**Figure 4 materials-14-06493-f004:**
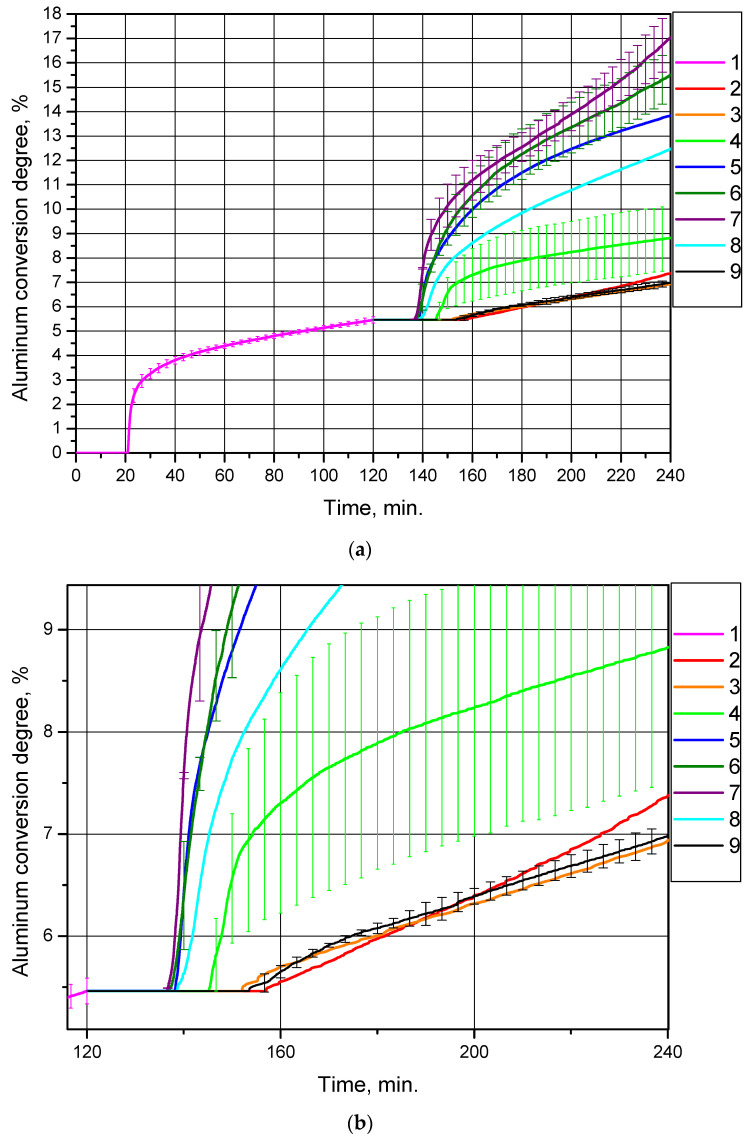
Aluminum conversion degrees (hydrogen yields) for different powder samples: (**a**) original size; (**b**) magnified section: 1—original aluminum powder, 2—sample after drying at 120 °C, 3—sample after thermal treatment at 200 °C, 4—sample after thermal treatment at 300 °C, 5—sample after thermal treatment at 450 °C, 6—sample after thermal treatment at 400 °C, 7—sample after thermal treatment at 500 °C, 8—sample after thermal treatment at 550 °C, 9—sample after thermal treatment at 600 °C.

**Figure 5 materials-14-06493-f005:**
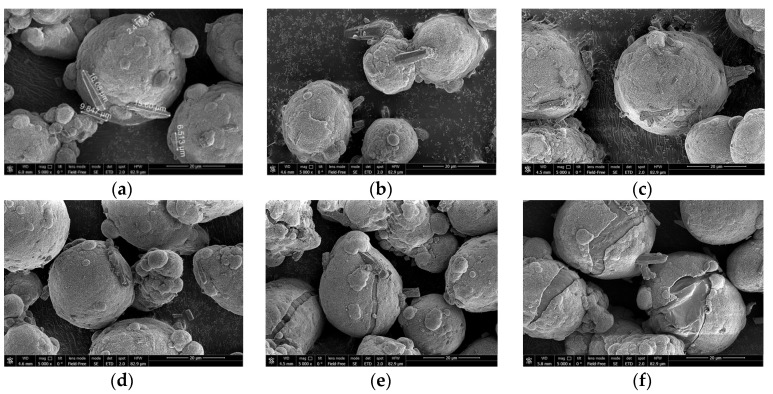
Micrographs of different samples of partially oxidized aluminum powder: (**a**) sample after drying at 120 °C; (**b**) sample after thermal treatment at 200 °C; (**c**) sample after thermal treatment at 300 °C; (**d**) sample after thermal treatment at 400 °C; (**e**) sample after thermal treatment at 500 °C; (**f**) sample after thermal treatment at 600 °C.

**Figure 6 materials-14-06493-f006:**
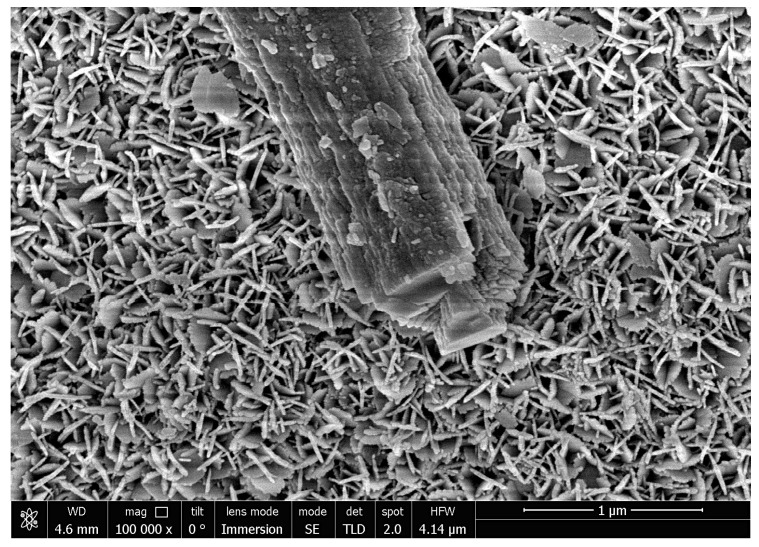
Micrograph of the reaction product layer for the sample thermally treated at 300 °C.

**Figure 7 materials-14-06493-f007:**
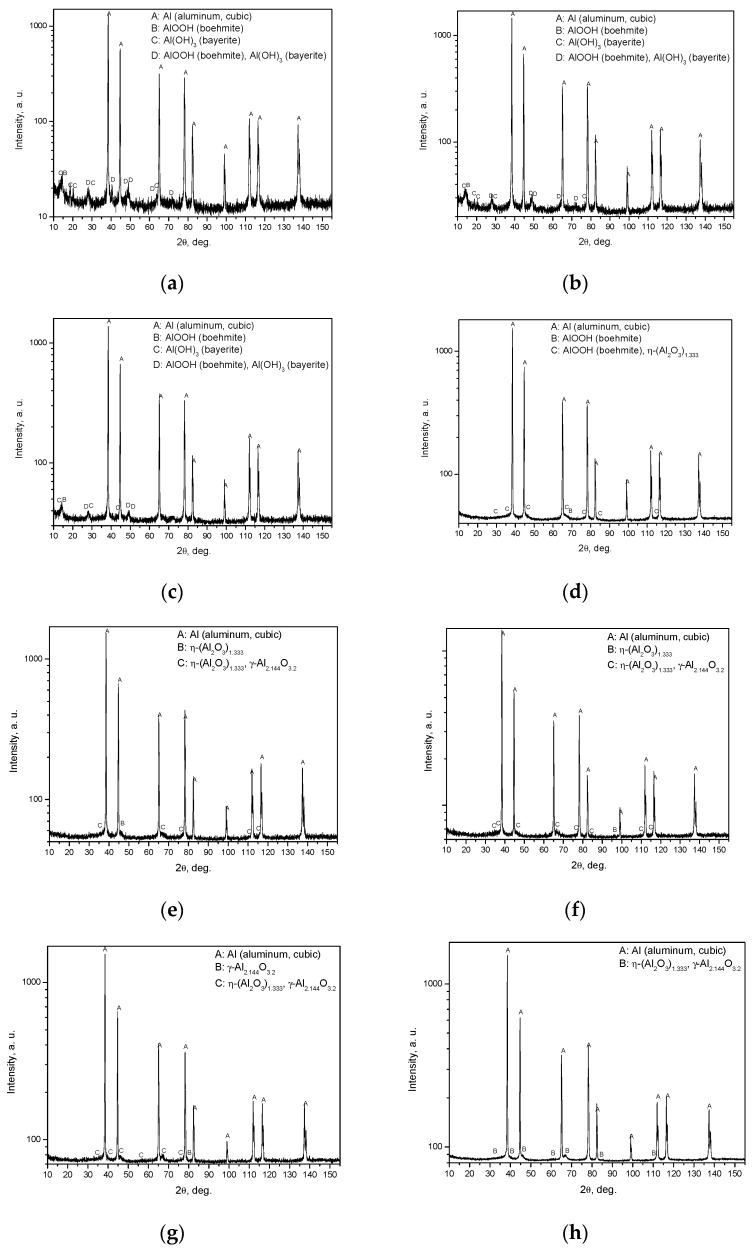
Results of X-ray diffraction analysis for different samples: (**a**) sample after drying at 120 °C; (**b**) sample after thermal treatment at 200 °C; (**c**) sample after thermal treatment at 300 °C; (**d**) sample after thermal treatment at 400 °C; (**e**) sample after thermal treatment at 450 °C; (**f**) sample after thermal treatment at 500 °C; (**g**) sample after thermal treatment at 550 °C; (**h**) sample after thermal treatment at 600 °C.

**Figure 8 materials-14-06493-f008:**
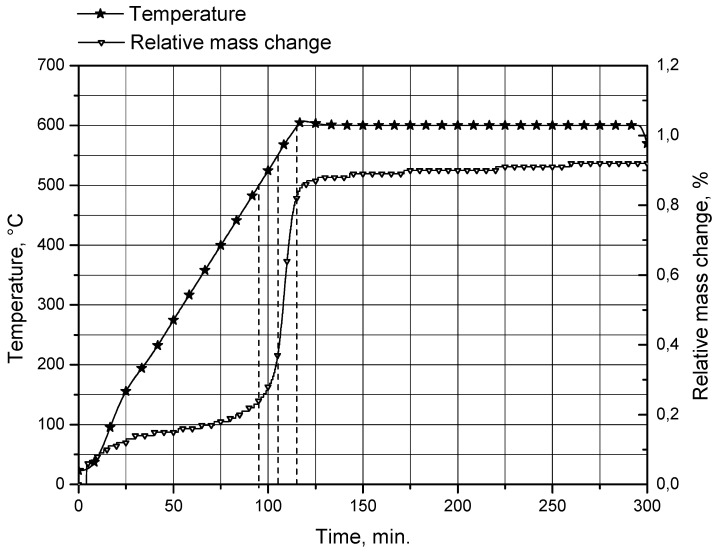
TG curve for original aluminum powder.

**Figure 9 materials-14-06493-f009:**
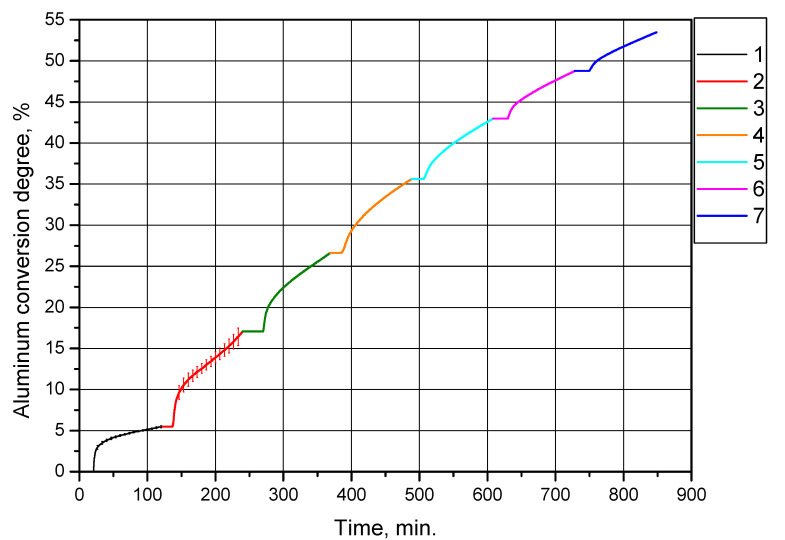
Aluminum conversion degrees (hydrogen yields) for the powder sample subjected to alternating oxidation and thermal treatment at 500 °C (with preliminary drying at 120 °C): 1—original aluminum powder, 2—sample after the first thermal treatment, 3—sample after the second thermal treatment, 4—sample after the third thermal treatment, 5—sample after the fourth thermal treatment, 6—sample after the fifth thermal treatment, 7—sample after the sixth thermal treatment.

**Table 1 materials-14-06493-t001:** Aluminum powder composition.

Element	Content (ppm)	Element	Content (ppm)
Al	base	Ga	90.7
Li	1.4	Y	0.1
Be	0.28	Zr	1.5
B	13.5	Mo	0.41
Na	9.6	U	0.41
Mg	16.2	Sn	1.0
Ti	9.9	Ba	0.2
V	32.2	La	0.72
Cr	4.7	Ce	0.81
Mn	13.7	Pr	0.04
Fe	771	Nd	0.16
Co	0.57	Sm	0.02
Ni	15.9	Gd	0.03
Cu	0.57	W	0.2
Zn	80.7	Tl	0.08

**Table 2 materials-14-06493-t002:** Aluminum conversion degrees (hydrogen yields) for different samples.

Sample	Conversion Degree (Total), %	Conversion Degree ± Standard Deviation (Per Experiment), %	Standard Deviation Averaged over Measurement Time, %
Original aluminum powder	5.46	5.46 ± 0.13	0.13
Sample after drying at 120 °C	7.38	1.92	–
Sample after thermal treatment at 200 °C	6.93	1.47	–
Sample after thermal treatment at 300 °C	8.82	3.36 ± 1.33	1.18
Sample after thermal treatment at 400 °C	15.51	10.05 ± 0.99	0.91
Sample after thermal treatment at 450 °C	13.82	8.36	–
Sample after thermal treatment at 500 °C	17.05	11.59 ± 1.20	0.74
Sample after thermal treatment at 550 °C	12.46	7.00	–
Sample after thermal treatment at 600 °C	6.99	1.53 ± 0.10	0.08

**Table 3 materials-14-06493-t003:** Aluminum conversion degrees (hydrogen yield) for the powder sample subjected to alternating oxidation and thermal treatment at 500 °C (with preliminary drying at 120 °C).

Sample	Conversion Degree (Total), %	Conversion Degree (Per Experiment), %
Original aluminum powder	5.46	5.46 ± 0.13
Sample after the first thermal treatment	17.05	11.59 ± 1.20
Sample after the second thermal treatment	26.61	9.56
Sample after the third thermal treatment	35.62	9.01
Sample after the fourth thermal treatment	42.97	7.35
Sample after the fifth thermal treatment	48.77	5.80
Sample after the sixth thermal treatment	53.46	4.69

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
