# Peer review of "Effect of Thermal Treatment of Aluminum Core-Shell Particles on Their Oxidation Kinetics in Water for Hydrogen Production"

_materials, 2021, doi:10.3390/ma14216493_

Round 1
Reviewer 1 Report
Recommendation: Publish with major revisions as noted
Comments: Authors have conducted the study evaluating the thermal treatments at various temperatures for the Al core-shell particles. XRD, SEM, and TGA were evolved. However, there are some issues and questions that need to be addressed before my full recommendation of this work to be published in Materials.
1) In Figure 4, though labeled with different symbols, it is very difficult to tell the curve from each other, especially in Figure 4(b)., readers won’t be able to tell which curve is for 1, and which is for 6, and 7. For that in Figure 4(a), it is the same issue with those at conversion degrees between 5-8%.
2) For the data in Table 2 and Table 3, how large are the uncertainties for each of the values provided?
3) Authors claimed that samples treated at 400C and higher temperatures contain alumina using the XRD data. Additional evidence should be provided such as XPS or SAXS.
4) For the claim on lines 322-324. An XPS depth profiling should be performed to support the claim that oxide films will become too thick.
Author Response
Dear Reviewer, Greetings from the authors!
We were pleased to get your feedback on our manuscript ‘Effect of Thermal Treatment of Aluminum Core-Shell Particles on Their Oxidation Kinetics in Water for Hydrogen Production’.
First of all, we would like express our gratitude to you for your comments, in accordance with which the following improvements were made:
- kinetic curves in Fig. 4 are now given in color, that makes possible to differentiate closely positioned and intersecting curves clearly;
- available data on the preliminary experiments (for original powder and samples thermally treated at 300, 400, 500 and 600 °Ð¡) from one of our internal unclassified reports were used to calculate the respective average data values and standard deviations, the data in the manuscript (graphs, tables and numeric values in the text) were updated, and some comments on the obtained results on standard deviations were added;
- unfortunately, in our institution no equipment for XPS or SAXS analysis is available. However, in order to confirm the given XRD-results on aluminum oxide phases registered in the reaction products at temperatures of 400 °Ð¡ and higher, experimental and simulation results from other studies were referred to;
- as we do not have the equipment for XPS or SAXS analysis at hand, we added a figure representing the results of XPS analysis of the oxide film thickness on solid aluminum borrowed from a reliable source (Schmitz, C. Handbook of aluminium recycling; Vulkan-Verlag GmbH: 2006). Now we are waiting for reply from the publishing house (copyright holder) to get an official permission for using the figure in our paper. Although we could not perform XPS analysis, we used a thermogravimetric (TG) method to measure the relative mass change for an aluminum powder sample heated up to 600 °Ð¡. The steepest section of the obtained TG-curve locates between 500 and 600 °C that correlates with a rapid increase in oxide film thickness illustrated in the copyrighted figure.
Thank you very much!
Reviewer 2 Report
The paper is excellently written. Thermal treatment was followed. The abstract is adequately written. In conclusion, the results are highlighted. All measurements with results and discussion were sent. The paper should be published in an unchanged edition.Author Response
Dear Reviewer, Greetings from the authors!
We were glad to get your feedback on our manuscript ‘Effect of Thermal Treatment of Aluminum Core-Shell Particles on Their Oxidation Kinetics in Water for Hydrogen Production’, and we are extremely pleased that our work received such a high evaluation.
Thank you very much!
Reviewer 3 Report
The paper is devoted to the ropical industrial problem (hydrogen production via the interaction of aluminum with the water)... In the paper, the new experimental aspects have been systematically studied, i.e. temperature treatment of the material, multiple cycles etc. From this point of view, the publication of the paper should be supported.
However, the main drawback of the paper is absence of theoretic explanation of obtained kinetic peculiarities presented in Fig.4 a and b (discontinuity, see 4a and hysteresis, see 4b)
Therefore, my decision is " To publish with major revision". The paper should be modified being added by the qualitative explanation at least.
Author Response
Dear Reviewer, Greetings from the authors!
We were pleased to get your feedback on our manuscript ‘Effect of Thermal Treatment of Aluminum Core-Shell Particles on Their Oxidation Kinetics in Water for Hydrogen Production’.
We considered your comments and made the following improvements to clarify the obtained results:
- kinetic curves in Fig. 4 (corrected in accordance to the comments of another reviewer) are now given in color, as the previous monochrome representation did not allow to differentiate closely positioned and intersecting curves clearly. Now it can be seen, that there is no hysteresis in Fig. 4b, but just a number of kinetic curves for different samples that have some intersections with each other at the region corresponding to the very beginning of reaction and have negligible effect on the actual trends;
- as to ‘disruptions’ in Fig. 4a, the first curve (closest to zero point) actually represents a kinetic curve for the sample of original aluminum powder, while the ‘bunch’ of ‘dendritic’ curves represents kinetic curves for the samples obtained from that ‘original’ sample which was reacting with water for 2 hours and then dried in a drying oven: after all, that sample was divided into 8 ‘subsamples’ with approximately equal masses, each of which was thermally treated in an individual manner (at 120, 200, 300, 400, 450, 500, 550 and 600 °C) and then again put into water to measure hydrogen yield. So, the ‘bunch’ is ‘attached’ to the end of the ‘first’ curve in order to illustrate a decrease in the total amount of unreacted aluminum in each sample: it decreased in the experiment with original aluminum powder and in experiment with each of thermally treated ‘subsamples’ obtained from that original aluminum powder which in its turn already had a reduced amount of unreacted aluminum after its reaction with water). Additional comments were added to the corresponding section of the text for the sake of more clarity.
Thank you very much!
Reviewer 4 Report
The manuscript "Effect of Thermal Treatment of Aluminum Core-Shell Particles on Their Oxidation Kinetics in Water for Hydrogen Production" is a well and carefully written work. There are a few minor aspects, which should be looked into, however. There are a few typos, but more importantly the authors should describe whether there were repeated tests (various parallels) or it was a single test per each condition. This aspect on statistics is not clearly reported. Also, standard deviations for results are not given and should be provided if possible.
All in all, in my opinion, the manuscript is a good work and deserves to be considered for publication. I attached a commented manuscript for your convenience.
//Accept after Minor Revision

Author Response
Dear Reviewer, Greetings from the authors!
We were glad to receive your positive feedback on our manuscript ‘Effect of Thermal Treatment of Aluminum Core-Shell Particles on Their Oxidation Kinetics in Water for Hydrogen Production’.
In accordance with your comments and suggestions we made the following revisions:
- missing citations were inserted;
- spelling mistakes, typing and translation errors were corrected;
- available data on the preliminary experiments (for original powder and samples thermally treated at 300, 400, 500 and 600 °Ð¡) from one of our internal unclassified reports were used to calculate the respective average data values and standard deviations, the data in the manuscript (graphs, tables and numeric values in the text) were updated, and some comments on the obtained results on standard deviations were added. As to the standard deviation for the set of seven successive tests – unfortunately, not all the experiments were repeated; however, the standard deviations for the first two experiments of the series were added.
Thank you very much for your time and attention to our work!
Round 2
Reviewer 1 Report
The authors have addressed the comments made in this updated version. I would recommend accept the manuscript in the current form.